# CP-Prompt: Composition-Based Cross-modal Prompting for Domain-Incremental Continual Learning

## ABSTRACT

The key challenge of cross-modal domain-incremental learning (DIL) is to enable the learning model to continuously learn from novel data with different feature distributions under the same task without forgetting old ones. However, existing top-performing methods still cause high forgetting rates, by lacking intra-domain knowledge extraction and inter-domain common prompting strategy. In this paper, we propose a simple yet effective framework, CP-Prompt, by training limited parameters to instruct a pre-trained model to learn new domains and avoid forgetting existing feature distributions. CP-Prompt captures intra-domain knowledge by compositionally inserting personalized prompts on multi-head self-attention layers and then learns the inter-domain knowledge with a common prompting strategy. CP-Prompt shows superiority compared with state-of-the-art baselines among three widely evaluated DIL tasks. The source code is available at https://anonymous.4open.science/r/CP_Prompt-C126.

## CCS CONCEPTS

• **Computing methodologies** → *Object recognition*.

## KEYWORDS

Prompts Learning, Cross-modal, Domain Incremental Learning

## 1 INTRODUCTION

Cross-modal models have garnered significant attention due to their capability to process and integrate diverse types of data. However, these models often encounter the challenge of different domains data feature distributions in practical applications. Domain Incremental Learning (DIL) [42] is a special incremental learning task, where the learning model is trained on a sequence of domains over time, with each domain or task presenting new and potentially information, e.g. distributional shift [1]. Under this setting, the tasks in each domain remain the same and the testing sample does not know which domain it belongs to. A vivid example is shown in Figure 1, where the learned model was firstly trained with qickdraw-style pictures, and then tested to classify the same category under different styles, such as infographics, painting, and clipart. The key success of DIL algorithm is to adapt and learn from sequential domains without forgetting the knowledge it has acquired from previous ones.

A key challenge for domain incremental learning is how to deal with the phenomenon of catastrophic forgetting [31, 32]. When learning new domains in sequence, the model may forget previous knowledge, leading to poor performance on old domains. To alleviate this issue, previous work [14, 35, 38] utilizes a buffer containing exemplars from previous tasks to facilitate learning new tasks. Then remarkable progress has recently been made in DIL tasks using prompt learning methods. Such as building prompt pool [46], adding different classification tokens [9], employing prompt on multi-modal pre-trained models [44].

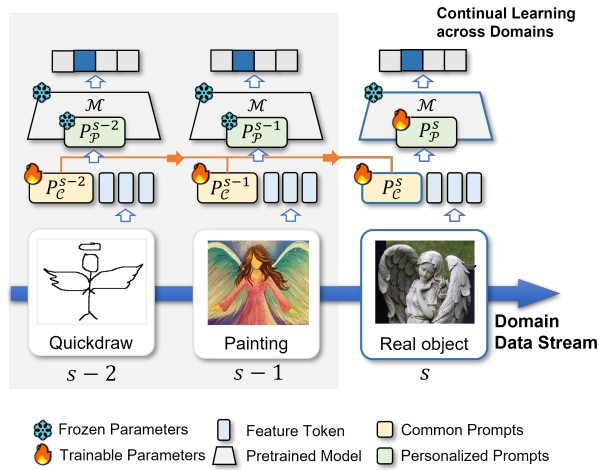

**Figure 1: A toy example of CP-Prompt in a domain-incremental learning task.**

Despite this, two challenges still remain: **(1) How to make the trade-off between common and personalized knowledge within DIL process?** Previous studies have shown that extracting common patterns between domains and enhancing personalized knowledge with each domain are both helpful in DIL. However, from the other side, how to balance inter-domain and intra-domain feature learning is still unaddressed. **(2) How to depict the impact of domain context on embedding tokens?** For the transformer which is widely adopted by DIL models, the effectiveness comes from routing information of lists of complex tokens to acquire the correlation by self-attention. However, this structure is difficult to learn information outside the fix-sized space of transformer [11]. Thus, additional domain context information should be guided into the transformer encoding process.

To this end, in this paper, we present a prompt learning framework, namely **CP-Prompt** (Common & Personalized), to instruct a pre-trained model to learn on incremental data domains with different data styles. As Figure 1 depicts, CP-Prompt adopts a twin-prompt strategy. The shared **common prompts**, embedding within shallow part of the model, are employed to learn knowledge of new domains sequentially and then frozen. Common prompts embedding of models can preserve knowledge among domains. The **personalized prompts**, called Prefix-One, embedded within the self-attention layers of the pre-trained model, contribute to model inference with domain style features. By incorporating these two prompts, the pre-trained model can be continually learned without tuning its original parameters.

The contributions of this paper are summarized as follows:

- We present a simple yet effective prompt tuning framework CP-Prompt for cross-modal domain-incremental learning, with a parameter-efficient twin-prompting design that preserved both

inter-domain common knowledge and intra-domain personalized knowledge.

- We further designed Prefix-One, which can incorporate domain context information into the self-attention layer, therey guiding the transformer to fully utilize domain knowledge at different semantic levels for DIL process.

- CP-Prompt is evaluated on three widely used DIL benchmark datasets and outperforms existing state-of-the-art sample-free baselines. Furthermore, only minimal additional parameters (0.22%) are tuned by CP-Prompt, and gaining at even 2.3% improvement, showing its effectiveness in both parameter efficiency and model accuracy.

## 2 RELATED WORK

*Domain Incremental Learning.* DIL refers to a type of continuous learning scenario where the feature distribution of the same task changes across different domains [42]. In other words, the data in each domain is used to accomplish the same task but differs from each other significantly [20, 33]. The goal of DIL is to enable the model to learn about newly added domains without retraining from scratch while maintaining its generalization in the original domains. Traditionally employed methods typically include architecture-, regularization-, and replay-based approaches. Architecture-based methods create independent components for each task or focus on task-specific subnetworks to avoid interference between network parameters to alleviate forgetting, such as XdG [30], DEN [48], PAE [17], and CPG [18]. Regularization-based approaches [21] [34]constrain or penalize significant model changes to keep memory on the previous domain with regularized losses such as distillation loss [25], and parameter update loss [50]. Replay-based methods mitigate catastrophic forgetting by preserving a small subset of the previous domain and replaying them when training new tasks, such as ER [39], DGR [2], iCaRL [38], BI-R [41], and A-GEM [6].

*Prompt Learning.* Prompt learning originated from manually designing templates as extra instructions to pre-trained models for efficient adaptation to downstream tasks. Compared with human-defined fixed ones, treating prompts as learnable parameters significantly enhances the efficiency and effectiveness of the model instruction [19, 29, 52]. In this setting, prompt learning only needs a tiny set of parameters for training instead of tuning the entire pre-trained model, benefiting much time- and cost-sensitive scenarios such as incremental learning and transfer learning [12, 13]. This parameter-efficient tuning method is primarily classified into three types, including addition-, specification, and reparameterization-based approaches. Addition-based methods introduce extra trainable neural modules not present in the original model, such as Prompt Tuning [22], Prefix Tuning [24], Adapter Tuning [15], and P-Tuning [26, 27]. Specification-based methods specify certain parameters in the original model as trainable and leave the rest frozen, such as BitFit [49]. Reparameterization-based methods transform existing parameters into a more parameter-efficient form, including LoRA [16], AdaLoRA [51], and QLoRA [8].

*Prompt Learning for DIL.* Compared with traditional approaches, prompt learning-based DIL methods have shown prominent advantages in both model performance and efficiency. For example, Dy-Tox [9] applies the vision transformer in continual computer vision tasks, eliminating the class token and devising personalized tokens for each task. L2P [46] employs a learnable key/prompt mechanism to select prompts added into image tokens based on the similarity between the keys and tokens. It introduces the concept of a prompt pool, based on a key-value mechanism to learn specific prompts within domains and make inferences by selecting appropriate prompts from the pool. Recent approaches such as S-liPrompts [44] achieve remarkable results which independently learn a set of prompts for each domain using prompt tuning only, but still overlooking shared knowledge between domains and leaving significant room for improvement in intra-domain training methods. Not only limited ti DIL tasks, the class incremental learning (CIL) methods also employ prompt to optimize models. Dual-Prompt [45] incorporates both general and expert prompts embedded in the pre-trained model, aimed at preserving personalized knowledge for retaining global shared knowledge and category distribution. HiDe-Prompt [43] expands the distance of category distribution using a contrastive loss penalty term, and identifies different tasks by optimizing the output layers.

## 3 PRELIMINARY

*Prompt Learning on Pre-trained Models.* Pre-trained models follow a paradigm that trains its parameters via massive self-supervised labeled data for general ability and fine-tunes them with few labeled data for downstream tasks. Prompt learning provides a tiny-parameter-sized embedding to guide a model to generate better responses, thereby significantly reducing the resource burden of model training and tuning. Taking the visual-text pre-trained model CLIP as an example, it comprises a visual encoder and a text one. In the image encoder, an image $x \in \mathbb{R}^{H \times W \times C}$ in encoded as a sequence of vectors $x_{emb} \in \mathbb{R}^{E_I \times D}$ by the visual encoder, where $H, W$ represents the resolution of the original image, $C$ is the number of channels, $E_I$ is the feature size after convolution, and $D$ is the embedding dimension. To perform prompt tuning on CLIP, we can inject tiny-sized parameters into a pre-trained model and only train them to adapt to downstream tasks. To formalize, the vector of image samples $x_{emb}$ is concatenated with soft prompts $P \in \mathbb{R}^{L \times D}$,

$$x_p = [P, x_{emb}] \in \mathbb{R}^{(E_I+L) \times D}, \quad (1)$$

where $L$ is the prompts length. Discovering the best prompts involves picking specific tokens, which can be achieved through either manual exploration or non-gradient-based search techniques [40].

The vector $x_p$ is then encoded by transformer layers, resulting in a high-dimensional projection $x_h \in \mathbb{R}^{H_I \times D}$, where $H_I$ is the number of image features in high dimensional space. Similarly, in the text encoder, we encode words through vocabulary and positional embedding $t_{emb} \in \mathbb{R}^{E_T \times D}$, which after transformer encoding also yields $t_h \in \mathbb{R}^{H_T \times D}$, where $E_T$ is the feature size, and $H_T$ is the number of text features in high dimensional space. The CLIP model seeks the mapping relationship between the two high-dimensional embeddings $x_h$ and $t_h$ through a contrastive loss. When predicting new-coming data, all parameters of CLIP are frozen, and the loss

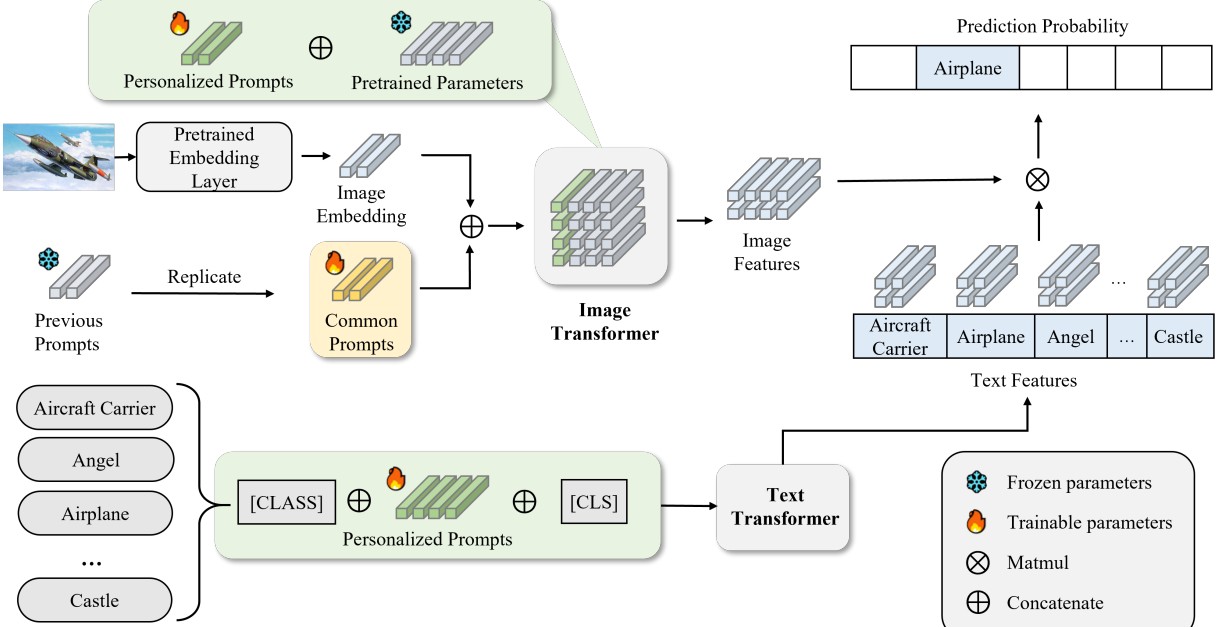

**Figure 2: The pipeline of CP-Prompt on new domain with twin-prompt structure. By taking CLIP as an example, the common prompts are sequentially trained based on the one from the previous domain, while domain-specific personalized prompts are embedded into key and value vectors to guide the model to learn the latent semantics. During the inference, similarity-based distances on embedding by $K$-Means determine the personalized domain prompts.**

gradient is only back-propagated to prompts parameters, thereby significantly reducing model training cost.

*Problem Definition.* In this paper, we take the widely-used multi-modal model-based image classification as the benchmark scenario [44]. Suppose there is a set of domains which each domain is denoted as $\mathcal{D}_s = \{x^{s,i}, y^{s,i}\}_{i=1}^N \in S$, where $x^{s,i} \in \mathbb{R}^{H \times W \times C}$ is the $i$-th image sample from the $s$ domain, and $N$ denotes the total number of samples. $y^{s,i}$ corresponds to the label associated with the sample. The feature distribution of data among different domains is highly heterogeneous, and each domain covers all the classes $U$ of the general task. In the DIL setting, only one domain data can be accessed by the learning model at one time. Additionally, data from previously visited domains cannot be used again when the model is training on the following domains. Formally, the learning model $\mathcal{M}$ begins training on $\mathcal{D}_1$ and progressively learns $\mathcal{D}_2, \ldots, \mathcal{D}_S$. The crucial challenge in DIL is to ensure that under $\mathcal{D}_S$, the model $\mathcal{M}$ can still maintain the performance on $\mathcal{D}_1, \ldots, \mathcal{D}_{S-1}$. The motivation of this research is to enhance the accuracy of tasks across all domains in this learning paradigm. Eventually, the objective of DIL is to optimize the following objective function:

$$\mathcal{L} = \sum_{i=1}^{s} \arg\min_{P} \mathcal{L}_i(F_{(P,\mathcal{M})}(x)) \tag{2}$$

where $F_{(P,\mathcal{M})}(x) \in \mathbb{R}^U$ is the prediction projection by model parameter $\mathcal{M}$ as well as tuned prompts parameters $P$, $\mathcal{L}_i$ is the prediction loss in $i$-th domain.

## 4 THE CP-PROMPT FRAMEWORK

### 4.1 Overall Structure

The overall pipeline of the proposed CP-Prompt is presented in Figure 2. In CP-Prompt, we propose a twin-prompting strategy. The underlying assumption of this design is that the learning model should be guided by inter-domain shared prompts to enhance the generalization of common knowledge for the overall task. Simultaneously, personalized prompts within the domain guide the model to capture personalized knowledge, improving accuracy for specificities. Specifically, personalized prompts are embedded into key and value vectors in different transformer layers for guiding the model to learn latent semantics with different granularities. During the inference, a simple $K$-Means algorithm is utilized to select appropriate common and personalized prompts to guide the pre-trained model to encode new image tokens for classification.

### 4.2 Common Prompts

As shown in Figure 2, we design a continually tuned common prompting strategy for guiding the learning model to extract shared knowledge across each domain. The common prompts are tiny-sized parameters that are tuned by loss gradient calculated by prediction on each domain data sample. At this moment, the entire parameters of the pre-trained model are frozen so that the generation variation of the model would only be affected by inputs and prompts.

For the sake of simplification, here we describe the prompting on the image encoder side. Formally, for the $i$-th domain dataset $\mathcal{D}_s = \{x^{s,i}, y^{s,i}\}_{i=1}^N$, the image tokens $x_{emb}^{s,i} \in \mathbb{R}^{E_I \times D}$ are obtained by the initial convolutional neural network embedding layer. Subsequently, $x_{emb}^{s,i}$ is concatenated with the common prompts $P_C^s \in \mathbb{R}^{L_C \times D}$.

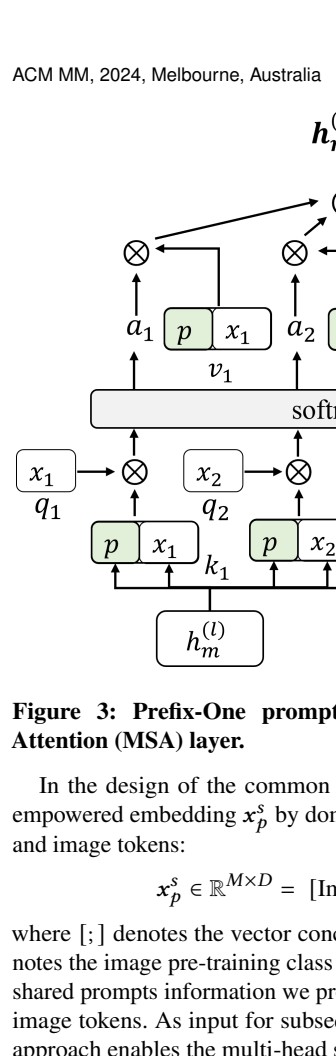

**Figure 3: Prefix-One prompting in the Multi-Head Self-Attention (MSA) layer.**

In the design of the common prompts, we composite prompt-empowered embedding $x_p^s$ by domain globally shared prompts $P_C^s$ and image tokens:

$$x_p^s \in \mathbb{R}^{M \times D} = [\text{Img}_{CLS}; x_{emb}^{s,i}; P_C^s], \qquad (3)$$

where $[;]$ denotes the vector concatenation operation. $\text{Img}_{CLS}$ denotes the image pre-training class tokens, and $M = E_I + L_C + 1$. The shared prompts information we propose are directly integrated with image tokens. As input for subsequent transformer encoding, this approach enables the multi-head self-attention (MSA) mechanism to effectively learn the prompt-guided token embeddings.

After the common prompts are tuned on one domain dataset, we make a copy of these prompts as frozen ones to serve as common prompts on domain $s$, and we sequentially tune the original prompts to fit the next domain $s + 1$, thereby obtaining $P_C^{s+1}$ sharing effective information and minimizing forgetting.

### 4.3 Personalized Prompts

In addition to the complementary formation of common prompts, personalized prompts are embedded across the transformer's attention layers in the form of parameters, capturing semantics at different granularities.

*Prefix-One Prompting.* The embeddings $x_p^s$ then undergo projection transformation by the transformer layers of the pre-trained model. Therefore, inserting personalized prompts $P_{\mathcal{P},img}^s \in \mathbb{R}^{L_{PI} \times D}$ into MSA layers can help to instruct the attention-capturing domain-individual semantics and knowledge, where $L_{PI}$ denotes the length of the personalized prompts.

Inspired by Prefix-Tuning [24], we incorporate prompts into the prefix section of data features. To apply domain context into the MSA layer, we add soft prompts to the *Key* matrix, and multiply it by the *Query* matrix containing only the original data, then derive attention score. The attention score calculates the original data and the contextual information related to the domain style. The attention

score is then multiplied by the *Value* matrix that is the same as the *Key*. The obtained output incorporates the contextual knowledge within the current domain, and this attention extension is named as Prefix-One, as presented in Figure 3. It enables the pre-trained model to consider the particularity of personalized domain style and achieve the effect of adapting to domain tasks with less training cost.

Formally, in Prefix-One within the MSA layers, we iteratively obtain encoding by compositing personalized prompts into key and value embeddings:

$$h_m^{(l+1)} = f_{Pre-One}^{(l)} \left( P_{\mathcal{P},img}^{(l)}, h_m^{(l)} \right)$$

$$= Softmax \left( \frac{h_q^{(l)} h_k^{(l)}}{\sqrt{d_{q,k}}} \right) \cdot h_v^{(l)}, \qquad (4)$$

where $l = 0, 1, \ldots R$ ($R$ indicates the number of transformer layers), $h_m^{(0)} = x_p^s$, and $h_q^{(l)}, h_k^{(l)}, h_v^{(l)}$ are calculated by:

$$h_q^{(l)} \in \mathbb{R}^{M \times D} = h_m^{(l)} W_q^{(l)}, \qquad (5)$$

$$h_k^{(l)} \in \mathbb{R}^{(M+L_{PI}) \times D} = [h_m^{(l)}; P_{\mathcal{P},img}^{(l)}] W_k^{(l)}, \qquad (6)$$

$$h_v^{(l)} \in \mathbb{R}^{(M+L_{PI}) \times D} = [h_m^{(l)}; P_{\mathcal{P},img}^{(l)}] W_v^{(l)}, \qquad (7)$$

where $P_{\mathcal{P},img}^{(l)}$ represents the image personalized prompts parameters for the $l$-th layer. $h_q^{(l)}, h_k^{(l)}, h_v^{(l)}$ are outputs to the $l$-th MSA layer in the image transformer. $W_q^{(l)}, W_k^{(l)}, W_v^{(l)}$ represent the corresponding model parameters. Since the prompts on the image side are embedded in the MSA layer, for a pre-trained model with $R$ layers of the transformer architecture, prompts can be embedded in multiple layers of MSA to better learn domain-specific knowledge.

*Generalizing to text encoder.* The Prefix-One for text-based encoder is similar to the image one. Taking our adopted CLIP architecture as an example, the complete label set $Y = \{y_j\}_{j=1}^U$ is transformed to texts. The encoded label text set $Y_{emb} = \{y_{emb}^j\} \in \mathbb{R}^{U \times D}$ is concatenated with the text personalized prompts $P_{\mathcal{P},tex}^s \in \mathbb{R}^{L_{PT} \times D}$, where $L_{PT}$ is the length of the personalized prompt for text. The high-dimensional projection after feature extraction by the text transformer is $t_h^s \in \mathbb{R}^{U \times D}$, and $L_T$ represents the feature count after encoding the label set using the vocabulary and adding positional vectors.

The composition of $P_{\mathcal{P},tex}^s$ is similar to the shared prompts on the image side. The encoded label $y_{emb}$ is concatenated with the class token $\text{Tex}_{CLS}$ and prompts $P_{\mathcal{P},tex}^s$, to derive $e^s$:

$$e^s \in \mathbb{R}^{(1+U+L_{PT}) \times D} = [\text{Tex}_{CLS}; P_{\mathcal{P},tex}^s; Y_{emb}]. \qquad (8)$$

After feature extraction by the frozen pre-trained transformer's multi-head attention mechanism, the classification token $t_h^s$ is obtained by:

$$t_h^s = g_{\mathcal{M}}(e^s \cdot W^s + b^s), \qquad (9)$$

where $W^s$ denotes the fixed parameters of the text pre-trained model, and $b^s$ represents the associated bias values, and the rest of this structure for text is the same as image one.

| Method | Buffer size | Prompt | AA | AF |
|---|---|---|---|---|
| LRCIL | | × | 76.39* | -4.39* |
| iCaRL | 100/class | × | 79.76* | -8.73* |
| LUCIR | | × | 82.53* | -5.34* |
| LRCIL | | × | 74.01* | -8.62* |
| iCaRL | 50/class | × | 73.98* | -14.50* |
| LUCIR | | × | 80.77* | -7.85* |
| DyTox | | ✓ | 86.21* | -1.55* |
| EWC | | × | 50.59* | -42.62* |
| LwF | | × | 60.94* | -13.53* |
| DyTox | | ✓ | 51.27* | -45.85* |
| L2P | | ✓ | 61.28* | -9.23* |
| HiDe-Prompt | No Buffer | ✓ | 84.32 | -2.61 |
| S-liPrompts | | ✓ | 88.79 | -0.63 |
| Dual-Prompt | | ✓ | 92.51 | -0.76 |
| **CP-Prompt(Ours)** | | ✓ | **93.65** | **-0.25** |

**Table 1: DIL Results on CDDB-Hard. * represents the result is quoted from [44].**

## 4.4 Overall Objective for CP-Prompt

Finally, the logits $z^s \in \mathbb{R}^U$, are computed by matrix multiplication of the high-dimensional projections from the image and text sides:

$$z^s = h_m^{(R)} \cdot (t_h^s)^\top. \tag{10}$$

It should be noted that, after continual training on $s$-th domain, $P_C^s$ is not only used as common prompts for the current domain, but also the initialization for the next one. However, the deep domain-specific knowledge-oriented personalized prompts $P_{\mathcal{P}}^s$ are isolated and optimized in different domains.

During the inference stage, we adopt a simple yet effective unsupervised clustering, $K$-Means, as domain selector to assign model extracted features with $K$ domain centroids $\mathbb{F} = \{m_f^j\}_{j=1}^s$ as feature pool. Given a new arriving sample $x_{new}$, the domain selector selects the most relevant personalized prompts for inference by measuring the distance between $x_{new}$ and $\mathbb{F}$. Following the multimodal pre-training setting [44], the model inferences prediction $z^{s'}$ is derived by freezing prompts parameters and untuned pre-trained model:

$$z^{s'} = f_{Pre-One}(P_C^s, P_{\mathcal{P},img}^s) \times g_{M_{pre}}(P_{\mathcal{P},tex}^s). \tag{11}$$

The goal of the model is to optimize the following loss function by tuning the tiny-sized prompts parameters $P_C^s, P_{\mathcal{P},img}^s, P_{\mathcal{P},tex}^s$:

$$\mathcal{L} = -\frac{1}{2n} \sum_{i=1}^n y \log(z^{s'}) + (1-y) \log(1-z^{s'}). \tag{12}$$

The overall training process is formally described in Algorithm 1.

## 4.5 Model Analysis

We conduct model analysis to demonstrate the rationality behind the simple design of CP-Prompt. We will demonstrate the relationship with Dual-Prompt, a class-incremental learning model that utilizes a combination of double prompts. We extracted Dual Prompt core modules and transformed them for use in domain incremental tasks. First we will explain the difference between General-Prompt

| Method | Buffer size | Prompt | AA |
|---|---|---|---|
| ER | | × | 79.75±0.84* |
| GDumb | | × | 74.92±0.25* |
| BiC | 50/class | × | 79.28±0.30* |
| DER++ | | × | 79.70±0.44* |
| Co²L | | × | 79.75±0.84* |
| L2P | | × | 81.07±0.13* |
| EWC | | × | 74.82±0.60* |
| LwF | | × | 75.45±0.40* |
| L2P | | ✓ | 78.33±0.06* |
| HiDe-Prompt | No Buffer | ✓ | 80.81±0.76 |
| S-liPrompts | | ✓ | 87.07±0.65 |
| Dual-Prompt | | ✓ | 88.74±0.36 |
| **CP-Prompt(Ours)** | | ✓ | **90.67±0.55** |

**Table 2: DIL Results on CORe50. * represents the result is quoted from [44].**

---

**Algorithm 1** The training algorithm of the CP-Prompt.

---

**Input:** $\mathcal{T}$: Pre-trained image model; $\mathcal{G}$: Pre-trained language model; $\mathcal{D}_s = \{x^{s,i}, y^{s,i}\}_{i=1}^N$: Training data; $Y_{emb} = \{y_{emb}^j\}$: Text class embeddings.

1: Initialization: Common Image Prompt Pool: $P_C$; Personalized Image Prompt Pool: $P_{\mathcal{P},img}$; Personalized Language Prompt Pool: $P_{\mathcal{P},tex}$; Domain centroids: $\mathbb{F}$
2: **for** $s = 1, 2, ...S$ **do**
3:    Initialize common prompt $P_C^s = P_C^{s-1}$ for domain s
4:    Initialize personalized image prompt $P_{\mathcal{P},img}^s$ for domain s
5:    Initialize personalized language prompt $P_{\mathcal{P},tex}^s$ for domain s
6:    $x_p^s$ Propagate by Eq. (3)
7:    $h_p^{(R)}$ Propagate by Eq. (4)
8:    $t_h^s$ Propagate by Eq. (8) and Eq. (9)
9:    Calculate class embeddings $Y_{emb}$ by pre-trained model $\mathcal{G}$
10:   Compute the prediction probability by Eq. (11)
11:   Compute the Cross-Entropy loss by Eq. (12)
12:   Update $\theta_{P_C^s}; \theta_{P_{\mathcal{P},img}^s}; \theta_{P_{\mathcal{P},img}^s}$
13: **end for**
14: Calculate training sample features by $\mathcal{T}(x^{s,i})$
15: Calculate domain clustering centroids $\{m_f^j\}_{j=1}^s$ by K-Means
16: Save centroids in the list $\mathbb{F}$

---

and Common-Prompt, and then demonstrate the difference between Expert-Prompt and Personalized-Prompt.

*Common prompts vs. General prompts.* In CP-Prompt, we design the common prompt as: $x_p = [P, x_{emb}] \in \mathbb{R}^{(E_I+L)\times D}$, where $x_p$ is the feature after initial encoding and $L$ is the prompts length. The purpose of this design is to retain shared information based on the characteristics of different data domains, and on the other hand, to enable multi-layer personalized prompts to deeply extract shared information. $g^{(l)} \in \mathbb{R}^{L_g \times D}$ is General prompts to be attached to the $l$-th MSA layers.

We explored embedding the General prompt in MSA layers, as shown in the figure 5. As the number of embedding layers increases,

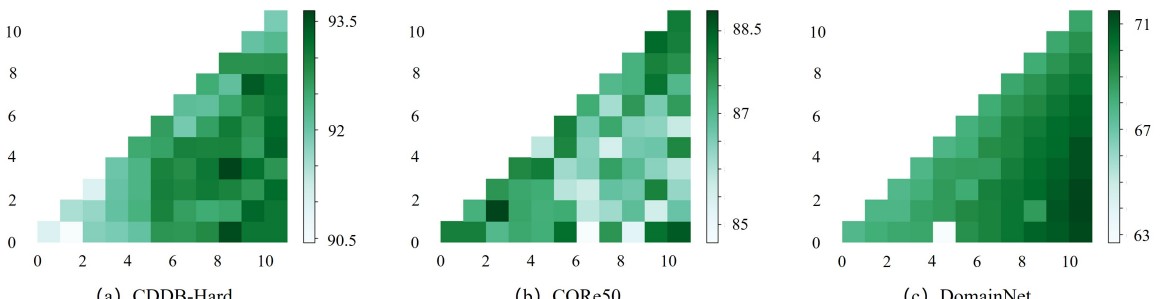

**Figure 4: The model performance variation results from inserting personalized prompts into different consecutive transformer layers. The vertical axis represents the starting layer index for inserting personalized prompts, while the horizontal axis represents the ending layer index.**

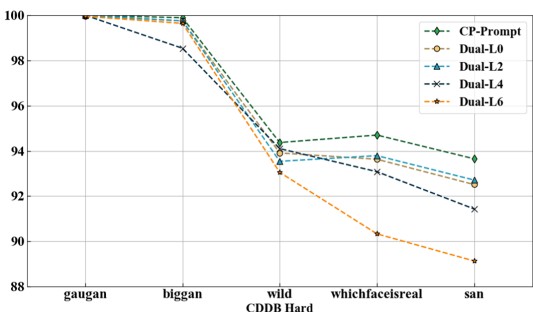

**Figure 5: Common prompts Comparison of different MSA layers embedding General prompts. The vertical axis is the average accuracy of the model. The horizontal axis from left to right is the domain data sequentially learned by the model.**

the forgetting rate of the model will increase significantly. The performance of our proposed method is better than the shallowest embedding of General Prompt.

Furthermore, targeting at the domain-incremental scenario, category information in complete before the dataset transfer, while the core challenge is to model the different feature distribution. Thus, the class-incremental-oriented methods such as DualPrompt fails to capture both the category-complete information and distribution difference. In the CP-Prompt, we propose to insert common prompts before the transformer structure is better than the design in the transformer, which the shared common prompts are employed to learn knowledge of new domains sequentially and then frozen.

*Personalized prompts vs. Expert prompts.* In CP-Prompt, personalized prompt considers the relationship between attention score and prompts, which the structure is formulated as: $f_{prompt}^{Pre-One}(p, h) = MSA(h_m, [p_k; h_k], [p_k; h_k])$. However, Dual-Prompt[45] splits $p$ into $p_k, p_v \in \mathbb{R}^{(L_p/2 \times D)}$, and connect them to $h_k, h_v$ respectively. The method of directly adding parameters is certainly effective, but in the attention layer, there is a lack of correlation between prompts.

In DIL scenarios, the parameters in the pre-training model are fixed. We argue that the attention score for the prompts should be further weighted, to learn the relationship between pre-training knowledge and domain-specific one, rather than simply adding parameters to fit new domain knowledge. As shown in Figure 5(b) of

| Method | Buffer size | Prompt | AA |
|---|---|---|---|
| DyTox | 50/class | ✓ | 62.94* |
| EWC | | ✗ | 47.62* |
| LwF | | ✗ | 49.19* |
| SimCLR | | ✗ | 44.2* |
| BYOL | | ✗ | 49.7* |
| Barlow Twins | No Buffer | ✗ | 48.9* |
| SupCon | | ✗ | 50.9* |
| HiDe-Prompt | | ✓ | 60.15 |
| S-liPrompts | | ✓ | 67.78 |
| Dual-Prompt | | ✓ | 71.02 |
| **CP-Prompt(Ours)** | | ✓ | **73.35** |

**Table 3: DIL Results on DomainNet. * represents the result is quoted from [44].**

the original manuscript, we compare the proposed tuning strategy, prefix-one, with the prefix-tuning under the same environment setting among the three benchmarks. The result shows that CP-Prompt achieves optimal performance, with a 2% improvement.

## 5 EXPERIMENT

### 5.1 Experiment Setup

*Dataset and Model Setting.* To evaluate the effectiveness of CP-Prompt, we test three widely used DIL benchmarks, including CDDB-Hard [23], CORe50 [36], and DomainNet [28]. For a fair performance comparison, we adopt the same dataset and experiment setting with the previous studies [44]. A detailed description of datasets and settings is available in supplementary materials.

*Baselines.* We compare the proposed CP-Prompt with state-of-the-art DIL methods, including replay-based methods including iCaRL [38], LUCIR [14], LRCIL [35], distillation-based method BiC [47], regularization-based method EWC [21], self-supervised-based method CaSSLe [10], and other non-prompt methods including ER [7], GDumb [37], DER++ [3], and Co$^2$L [5]. Furthermore, prompting-based methods including L2P [46], DyTox [9] and S-liPrompts [44] are also compared. In addition, we also extend two class incremental learning methods, includung Dual-Prompt [45] and HiDe-Prompt [43], to the DIL task as baselines. A detailed

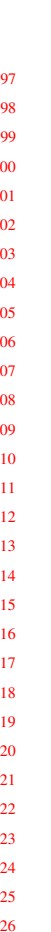

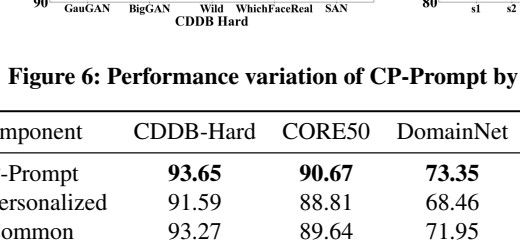

**Figure 6: Performance variation of CP-Prompt by (a) using different prompt lengths; (b) adding new domain data.**

| Component | CDDB-Hard | CORE50 | DomainNet |
|---|---|---|---|
| CP-Prompt | **93.65** | **90.67** | **73.35** |
| -Personalized | 91.59 | 88.81 | 68.46 |
| -Common | 93.27 | 89.64 | 71.95 |
| -Both | 88.79 | 88.07 | 67.78 |

**Table 4: Ablation results of CP-Prompt on three datasets**

description of these baselines is also available in supplementary materials. For a fair model comparison, all methods utilizing pre-trained models are standardized to use the image transformer encoder and text transformer encoder from CLIP.

*Evaluation Metrics.* We employ widely used two standard metrics in DIL, average classification accuracy (AA) and forgetting rate (AF) as the evaluation metrics for comparing the CP-Prompt and other baselines. Formally, let $A_{i,n}$ represent the model evaluation score (i.e., classification accuracy in our experiment) on the $i$-th domain after training on the $n$-th domain. AA and AF are measured as follows:

$$AA = \frac{1}{n} \sum_{i=1}^{n} A_{i,n}, \quad (13)$$

$$AF = \frac{1}{n-1} \sum_{i=1}^{n-1} \left( \frac{1}{n-i-1} \sum_{j=i+1}^{n} (A_{i,j} - A_{i,i}) \right). \quad (14)$$

It should be noted that AA evaluates the learning model's overall absolute performance, while AF measures the model's ability to overcome catastrophic forgetting. A successful DIL model should strive for high AA while maximizing AF towards 0.

## 5.2 Experimental Results

*Main Results.* Experimental results from Table 1, 2 and 3 demonstrate that our proposed CP-Prompt method significantly outperforms other exemplar-free methods, including the recently introduced state-of-the-art domain incremental method S-liPrompt. We

slightly extend the Dual-Prompt and HiDe-Prompt to adopt to the DIL setting. Both of them also take the same multi-modal pre-trained model CLIP. It is observed that the prompt design proposed by Dual-Prompt also shows a mild improvement. In contrast, adding HiDe-Prompt produces negative optimizations, showing that the idea of retaining finer-grained category features is not effective for DIL tasks. In particular, we achieved the optimal average classification accuracy in 2-class, 50-class, and 345-class DIL tasks, with the highest improvement reaching 2.32%. Additionally, we attained the optimal average forgetting rate, reduced to 0.25.

Compared to traditional historical replay-based methods, our proposed prompting approach significantly enhances classification performance for data with more similar features without using an additional sample buffer. In fact, sampling a small amount of information from the original domain is likely to introduce extra noise and, consequently, fails to improve classification performance.

In contrast to the other approach of prompt methods, CP-Prompt incorporates domain-wide shared prompts to facilitate the transfer of common knowledge. Additionally, our approach employs a multi-layered intra-domain prompting method, making full use of self-attention mechanisms to merge prompt information with high-dimensional latent features. The combination of these two prompts optimally leverages shared information across domains and individualized information within each domain. Compared to the SOTA model, our proposed method even surpasses the upper limit of S-liPrompts on i.i.d. data in terms of the CDDB-Hard and DomainNet tasks. This observation indicates that our proposed method is more effective in handling data with similar categories and significant feature differences in highly heterogeneous domains compared to fine-tuning methods.

*Ablation Study.* We also perform the ablation study to evaluate the effectiveness of each major component in CP-Prompt, as shown in Table 4. We evaluate the performance by removing personalized,

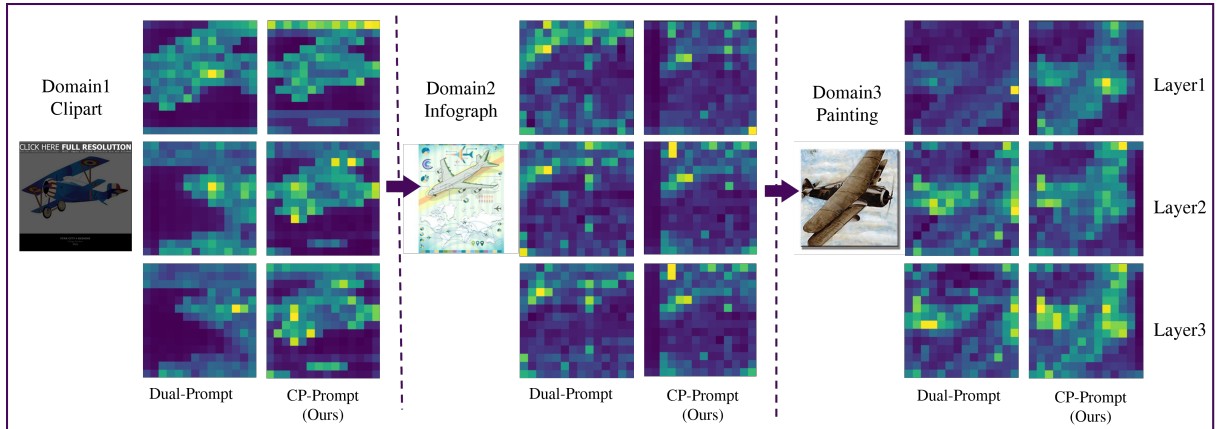

**Figure 7: Attention weights (layer 1 - 3) of CP-prompt and Dual-Prompt when shifting on different domains.**

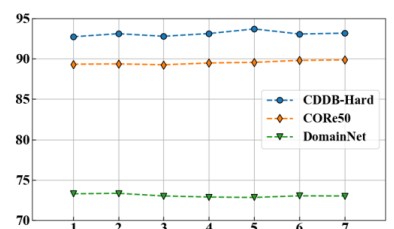

**Figure 8: Different $K$ values for $K$-Means.**

common, and both prompting, which is denoted as '-Personalized', '-Common', and '-Both', respectively. It is observed that both common and personalized prompts play a crucial role in improving performance for DIL tasks. Removing the common prompts leads to a decline in performance across the three tasks, as inter-domain knowledge sharing becomes impaired. Similarly, omitting the personalized prompts results in an ineffective extraction of knowledge within domains, leading to substantial performance loss.

*Model Parameter Analysis.* We further explore the optimal layers for inserting domain prompts, as illustrated in Figure 6. We observe that inserting prompts is widespreadly effective for transformer layers. Additionally, for new arriving DIL samples from known domains (CDDB-Hard and DomainNet), prompts within deep layers contribute to the performance more significantly by extracting high-level individual information. In the case of a sample from unknown domains (CORe50), adding prompts in both the shallow and deep layers helps to guide the model to capture the common and personalized knowledge of domains.

Prompt length is a hyper-parameter of CP-Prompt. As shown in Figure 6-(a), the model's performance is generally insensitive to the length of common prompts. However, an excessively long common prompt may introduce information specific to certain domains, leading to a slight performance decline. In the case of personalized prompts, adopting a longer prompt length than common prompts can significantly improve model accuracy to gain a larger encoding space. However, compared with fine-tuning, our method has a total parameter count of 150 million, with the actual fine-tuned parameters being 335,360, accounting for 0.22% in each domain.

In Figure 6-(b), we compare the improvement in Prefix-One design with existing methods, namely S-liPrompts and Prefix Tuning. The results indicate that our proposed solution exhibits superior performance in each domain. Moreover, in domains with lower data quality, CP-Prompt demonstrates lower forgetting rates, leading to overall better performance. In Figure 8, we explore the effects of the number of clustering points in each domain. The results indicate that increasing the number of clustering points generally leads to an imperceptible performance improvement. Thus, the choice of $K$ is not a crucial setting of CP-Prompt.

*Common Prompt Design Analysis.* We further explore contrasting different prompt design schemes between Dual-Prompt and the propsoed CP-Prompt. As introduced above, Dual-Prompt embeds the General Prompt within the attention layer, while CP-Prompt embeds the Common Prompt before the attention layer. As shown in Figure 7, we adopted the Dino visualization scheme [4] to map the attention variation when the model continues to learn on different domains. The attention weights of the first three layers of both approaches are displayed. It is observed that the attentions of CP-Prompt keep close to the object for identification alone side with domain shifting, while the Dual-Prompt gradually lose correct focus. It is evident that for cross-domain learning processes, embedding multi-domain common knowledge before attention better preserves key information in each domain. Particularly in info-graphs with multiple informational elements, the CP-Prompt approach can more accurately identify primary characteristic information.

## 6 CONCLUSION

In this paper, we propose CP-Prompt, which introduces common and personalized prompts into the cross-modal domain-incremental learning task. CP-Prompt integrates prompts into pre-trained models based on the transformer architecture, learning common prompts in the shallow layers and personalized prompts in the deep layers to capture semantic knowledge at different granularity. CP-Prompt significantly reduces the catastrophic forgetting rate by only tuning tiny-sized parameters. Extensive experiments also show the superiority of CP-Prompt over existing state-of-the-art approaches.

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
