# OpenReview forum: "CP-Prompt: Composition-Based Cross-modal Prompting for Domain-Incremental Continual Learning"
_acmmm.org/ACMMM/2024/Conference — MM2024 Poster_

### Official Review · Reviewer_6QJp · 2024-05-14

**Rating:** 4
**Confidence:** 3

**Summary:**

This paper addresses the challenge of cross-modal domain-incremental learning (DIL), which involves enabling a learning model to continuously learn from novel data with varying feature distributions under the same task without forgetting previous knowledge. Existing top-performing methods still result in high forgetting rates due to the lack of intra-domain knowledge extraction and an inter-domain common prompting strategy.
The authors propose a framework named CP-Prompt, which is a simple yet effective approach that trains a limited number of parameters to guide a pre-trained model to learn new domains and prevent forgetting of existing feature distributions. CP-Prompt captures intra-domain knowledge by compositionally inserting personalized prompts into the multi-head self-attention layers of a pre-trained model. It then learns inter-domain knowledge with a common prompting strategy. The framework is evaluated over three DIL tasks.

**Strengths:**

1. The paper deals with an important problem which is definitly relavent to the scope of the venue.
2. The paper is clearly written and the method description is easy to follow.
3. The proposed method demonstrate its effectiveness both on the DIL tasks and on the parameter efficiency.

**Limitations:**

1. The model parameter analysis part claims an accounting of 0.22% fine-tued parameters. However, the comparison on the parameter counting to other methods is missing.
2. The evaluation section proposes two metrics, namely AA and AF, however, the result of AF is missing in the ablation study.
3. Minor suggestions:
 Line 207 in -> is
 Tables: You may add upper/down arrows after the metrics to indicate the desired values.
 Line 553-556, these presentations are not proper here.

**Suitability:**

3

---

### Official Review · Reviewer_vewY · 2024-05-15

**Rating:** 5
**Confidence:** 3

**Summary:**

This paper proposes a simple and effective framework, CP-Prompt, to instruct the pre-trained model to learn new domains by training a limited number of parameters and avoid forgetting the existing feature distributions.

**Strengths:**

CP-Prompt captures domain knowledge by inserting personalized prompts on the multi-head self-attention layer, and then learns inter-domain knowledge with a common prompting strategy. The experimental results are better than baselines. Therefore, the proposed method are effective.

**Limitations:**

(1) Although this paper proposes a domain incremental learning model based on Prompt, it is weak in theoretical derivation and proof. It is suggested that the theoretical part should be enhanced to explain in detail the theoretical basis of the model design and how the proposed method can effectively control the interaction between different prompts.
(2) Although the model performance was demonstrated on several standard datasets, there was a lack of deep analysis of the model's generalization ability. It is recommended to add relevant generalization experiments to demonstrate the robustness of the model under various conditions.
(3) Grammatical and spelling errors: 43 lines of "qickdraw" should be changed to "quick draw".

**Suitability:**

3

---

### Official Review · Reviewer_AMav · 2024-05-28

**Rating:** 3
**Confidence:** 3

**Summary:**

This paper introduces a new prompt learning framework named CP-Prompt for cross-modal domain-incremental learning, which effectively instruct a pre-tained model to learn new domains and avoid forgetting existing feature distributions.

The main contribution of this method is to to effectively incorporate a twin-prompt strategy into pre-trained models. The shared common prompts is used to learn the inter-domain knowledge and the personalized prompts capture intra-domain knowledge. This method uses K-Means algorrithm to select appropriate common and personalized prompts during the inference stage, which means there is no need to provide a task ID during inference.

The paper provides a comprehensive comparison of this method on three widely used domain-incremental learning benchmark. The results indicate that CP-Prompt, outperforms the compared methods in terms of both average classification accuracy and the ability to avoiding forgetting.

**Strengths:**

1. The method of this paper is well-written and easy to follow.

2. The reported results outperforms the compared methods in terms of both average classification accuracy and the ability to avoiding forgetting.

3. The paper includes sufficient experiments, and the visual analysis results clearly demonstrated the proposed method can more accurately identify primary characteristic information.

**Limitations:**

1. The paper lacks technical novelty and fails to attract attention in terms of technical innovation (using MSA for common prompt learning and prompt information interaction in both the Common Prompts and Personalized Prompts learning approaches). Furthermore, with the advancements in effective parameter fine-tuning methods like Lora and Adapters, using prompt learning as the primary approach for continual learning lacks domain advancement. As a comparative study, another paper [1] also utilizes prompt learning for knowledge transfer.

2. The overall framework diagram of the paper could simultaneously illustrate the process during the inference stage and highlight the prompt selection process.

3. Comparing the effects with a multi-task learning framework might better reflect the task's difficulty.

4. I'm curious whether experiments related to the sequential robustness are not necessary in this continual learning setting, as the experimental results do not clearly demonstrate it.

[1] Li Y, Yang X, Wang H, et al. Learning to Prompt Knowledge Transfer for Open-World Continual Learning[C]//Proceedings of the AAAI Conference on Artificial Intelligence. 2024, 38(12): 13700-13708.

** Possible typos **
579: figure -> Figure
637: Figure 5(b) -> Figure 6(b)?

Overall, this paper may not introduce strong technical novelty (With the proposal of more effective parameter fine-tuning methods such as Adapters and Lora, it seems a bit outdated to continue using prompt learning as the main approach to address forgetting.

Moreover, the use of K-means to find personalized prompt parameters (similar to finding task IDs) is not technically advanced. I am a bit confused about the fact that the clustering centers in K-means have little impact on the clustering results. Shouldn't the number of personalized prompts decrease as the number of clusters increases? Does the reduction in prompt parameter space not affect the inference results? I don't quite understand this), it offers a good empirical study of lightweight continual learning techniques.

**Suitability:**

3

---

### Meta-Review · Area_Chair_E52b · 2024-06-29

**Recommendation:** Accept (Poster)
**Confidence:** 4

**Metareview:**

This paper presents a new prompt learning framework named CP-Prompt for cross-modal domain-incremental learning, which effectively guides a pre-trained model to learn new domains while preserving existing feature distributions. The main contribution of this method lies in the integration of a twin-prompt strategy into pre-trained models. Shared common prompts are used to acquire inter-domain knowledge, while personalized prompts capture intra-domain knowledge. During inference, the method employs a K-Means algorithm to dynamically select appropriate common and personalized prompts, eliminating the need for providing a task ID. Evaluations on several widely used domain-incremental learning benchmarks demonstrate the effectiveness of the proposed method.

All reviewers found the paper interesting and well-written. The evaluations are comprehensive and the experimental results are sound. I would encourage the authors to incorporate the changes and discussions in their rebuttal into the final version.